# MEDEVAL: A Multi-Level, Multi-Task, and Multi-Domain Medical Benchmark for Language Model Evaluation

**Zexue He[◇], Yu Wang[◇], An Yan[◇], Yao Liu[◇], Eric Y. Chang[◇,♣],**
**Amilcare Gentili[◇,♣], Julian McAuley[◇], Chun-Nan Hsu[◇,♣,♡]**

[◇]University of California, San Diego, La Jolla, CA, United States
[♣]Veterans Affairs San Diego Healthcare System, San Diego, CA, United States
[♡] Veterans Affairs National Artificial Intelligence Institute, Washington, DC, United States
{zehe, yuw164, ayan, yal004, e8chang}@ucsd.edu
{agentili, jmcauley, chunnan}@ucsd.edu

## Abstract

Curated datasets for healthcare are often limited due to the need of human annotations from experts. In this paper, we present MEDEVAL, a multi-level, multi-task, and multi-domain medical benchmark to facilitate the development of language models for healthcare. MEDEVAL is comprehensive and consists of data from several healthcare systems and spans 35 human body regions from 8 examination modalities. With 22,779 collected sentences and 21,228 reports, we provide expert annotations at multiple levels, offering a granular potential usage of the data and supporting a wide range of tasks. Moreover, we systematically evaluated 10 generic and domain-specific language models under zero-shot and finetuning settings, from domain-adapted baselines in healthcare to general-purposed state-of-the-art large language models (e.g., ChatGPT). Our evaluations reveal varying effectiveness of the two categories of language models across different tasks, from which we notice the importance of instruction tuning for few-shot usage of large language models. Our investigation paves the way toward benchmarking language models for healthcare and provides valuable insights into the strengths and limitations of adopting large language models in medical domains, informing their practical applications and future advancements[1].

## 1 Introduction

Recent advanced language models, e.g., GPT-3, ChatGPT, and LLaMa (Touvron et al., 2023a), are effective in various general tasks, suggesting their potential to healthcare use cases, such as alleviating the burden on human experts in decision-making and patient care. However, training, adapt-

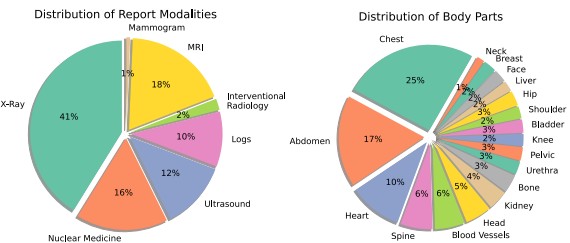

(a) Distribution of modality and body parts in MEDEVAL

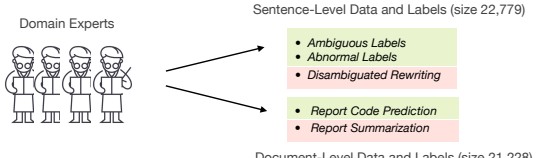

(b) Multi-task expert labels at multi-granularity

Figure 1: A summary of the multi-level multi-task and multi-domain medical benchmark (MEDEVAL). Classification tasks are highlighted in green and generation tasks are highlighted in red.

ing, and evaluating these models requires high-quality domain-specific datasets, which are often challenging to obtain. Previous medical datasets have been collected from healthcare-related literature (Dernoncourt and Lee, 2017; Gupta et al., 2021; Jin et al., 2019b; Banarescu et al., 2013) or web pages on the Internet (McCreery et al., 2020; Ammar et al., 2018). While these datasets are large, they may lack quality with heterogeneous topics (e.g., scientific literature about nutrition may offer limited help in the decision-making process of an X-ray analysis). On the other hand, high-quality clinical data is typically obtained by annotating records from healthcare systems like MIMIC-CXR (Johnson et al., 2019; Yan et al., 2021a). However, such data is either limited in size (Tsatsaronis et al., 2015), or may only cover certain dominant systems and specific domains[2] , such as chest X-rays (John-

---

[1]We will release the data set and source codes at https://github.com/ZexueHe/MedEval The data set will also be available at the Department of Veterans Affairs Open Data Portal https://www.data.va.gov

[2]We use *domain* to describe data sets that are different in distribution, caused by the modality of the examination (e.g.,

son et al., 2016) or eye diseases (Otmakhova et al., 2022). Other approaches involve automatically generating medical corpus using templates (Pampari et al., 2018; Pappas et al., 2018) or using language models (Guo et al., 2023; Tang et al., 2023), but they have been noted to be limited in diversity, complexity, and quality (Gupta et al., 2021).

To tackle the aforementioned challenges and facilitate research in clinical NLP, we introduce MEDEVAL, a large-scale medical benchmark with multi-level curated labels for multiple tasks and multiple domains. MEDEVAL comprises 22,779 sentence-level datapoints from radiology reports, including expert-crafted classification labels (e.g., abnormality identification labels) and ground truth for generation tasks (e.g., disambiguated rewritings). Additionally, we include 21,228 complete reports with expert-annotated medical codes for disease classification (e.g., for ankle radiology studies) and golden output for generation tasks (e.g., summarization of radiology reports). Besides the ability to support multi-tasks at different levels, MEDEVAL's uniqueness also lies in its diverse data coverage for different body parts (such as chest, foot, and ankle) and different modalities (X-rays, ultrasound, etc.), and the incorporated novel tasks/data that are collected from the U.S. Department of Veterans Affairs (VA) health care system nationwide. To the best of our knowledge, MEDEVAL represents the first expert-curated medical NLP benchmark that is both comprehensive and large-scale. MEDEVAL will be released to facilitate future research.

We further conduct a comprehensive evaluation of multiple state-of-the-art language model baselines, including domain-adapted PLMs followed by in-domain fine-tuning (e.g., fine-tuned BERT (Devlin et al., 2018)) and general-purposed LLMs utilized with few-shot in-context learning (e.g., ChatGPT). We evaluate their performance on sentence-level and document-level NLU and NLG tasks. We observe the effectiveness of both categories of models in different healthcare tasks, with surprisingly comparable performances from LLMs only using few-shot learning to domain-adapted PLMs in certain generation tasks. Our comprehensive evaluation indicates language models are strong candidates in medical tasks whose data is already seen/similar to their training data. Our investigation provides insights into the potentials and

limitations of LLMs in healthcare domains, guiding the appropriate use of LLM-assisted healthcare decision-making systems in the future. Overall, our contributions are summarized as:

- We propose a large-scale medical benchmark, namely MEDEVAL, with a broad coverage for various tasks and domains to facilitate future research in clinical NLP.

- We provide expert annotations for multiple tasks with multi-granularity, from sentence classification and rewriting, to report classification and summarization.

- We systematically evaluate various language models, and shed light on the strengths and weaknesses of these models for healthcare applications.

## 2 Related Work

**Medical Benchmark** Existing medical benchmarks are typically collected from the following resources. First, data is crawled from public resources, such as biomedical journals and literature like PubMed Central (PMC) or Semantic Scholar Ammar et al. (2018), or healthcare-related web pages such as MQP (McCreery et al., 2020) which was proposed by collecting COVID-19 FAQs from the Internet. Those data are heterogeneous and may lack relevance for assisting certain clinical purpose.

On the other hand, high-quality benchmarks are extensively collected from real-world healthcare systems, such as the Medical Information Mart for Intensive Care databases (MIMIC (Johnson et al., 2016, 2019, 2023)). Several works have been proposed based on their databases. For instance, MIMIC-CXR (Johnson et al., 2019) is a dataset consisting of pairs of radiology images and reports of chest X-ray exams. MIMIC PERform Dataset (Charlton et al., 2022) comprises physiological signals related to critically-ill patients. Additionally, Edin et al. (2023) collected document summary pairs labeled with diagnosis and procedure codes from Johnson et al. (2023). Though their collection may be one of the most comprehensive, there are many domains not included. A more recent attempt was made to alleviate the incompleteness concern by introducing M3 (Otmakhova et al., 2022), a multi-domain medical benchmark that incorporates multi-level expert annotations. By only considering studies on ophthalmology, M3 offers limited help in providing a fully comprehensive solution.

---

X-ray, CT, Ultrasound, etc) and examined body parts (e.g., chest, abdomen, etc).

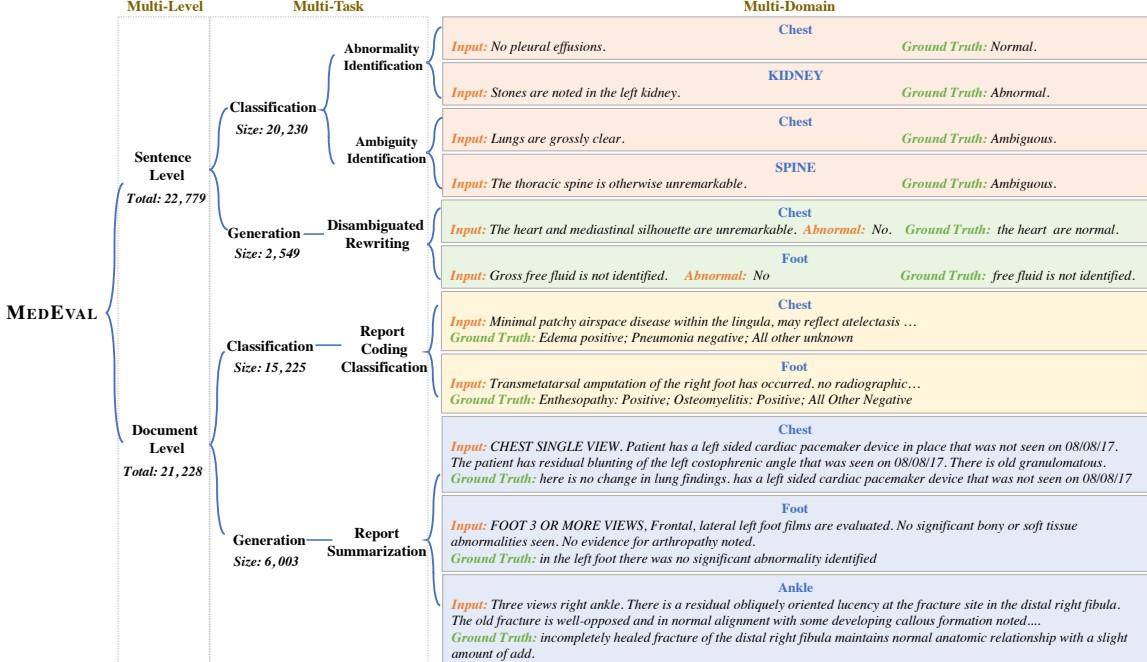

Figure 2: Dataset composition of MEDEVAL. MEDEVAL is a large-scale benchmark composed of 22,779 report sentences and 21,228 reports, covering multiple exam modalities on diverse body parts.

To complement these existing efforts, we collect a large-scale dataset of real medical reports from another healthcare system that offers broader coverage including 35 human body regions from 8 examination modalities (e.g., X-ray, CT, etc.).

**Language Models for Healthcare** Large pre-trained language models are being widely adopted to solve healthcare tasks. One line of research involves adapting general language models to the biomedical domain through continuous training on domain-specific data and tasks. For instance, Yan et al. (2021b) enhanced BERT with contrastive learning for chest report generation. ClinicBERT (Huang et al., 2019) was proposed by continuously training BERT on clinic notes using masked language modeling, and Yan et al. (2022) developed RadBERT by continuously training BERT on a vast collection of radiology reports. Other adaptations of BERT, such as BioBERT (Lee et al., 2020), Blue-BERT (Peng et al., 2019), SciBERT (Beltagy et al., 2019), and BioMegatron (Shin et al., 2020), involved training on large publicly available medical corpora like PubMed or Semantic Scholar. Furthermore, LLMs of alternative architectures have also been employed, including BioELMo (Jin et al., 2019a), BioBART (Yuan et al., 2022), and BioMed-RoBERTa (Gururangan et al., 2020a). Another research direction capitalizes on the generaliza-

tion capabilities of recent LLMs, where biomedical problems are addressed through prompting LLMs in zero-shot or few-shot settings. This approach has been utilized in various applications, such as medical report summarization (Otmakhova et al., 2022), medical writing (Biswas, 2023), and medical named entity recognition (Hu et al., 2023), etc.

In this work, we propose MEDEVAL, a multi-level data with curated annotations at various granularity to comprehensively evaluate the strengths and limitations of LMs in healthcare.

## 3 Dataset Design

MEDEVAL (shown in Figure 2) is designed with multiple NLU and NLG tasks at both the sentence and document levels, based on medical data collected from two different healthcare databases. Our data covers diverse combinations of human body parts and examination modalities. We first introduce the data sources where we collected the text input (Section 3.1). Then we present the expert-annotated ground truth labels[3] created by our medical team[4] (Section 3.2).

---

[3] We use "ground truth labels" to represent both discriminative labels for NLU and golden sentences for NLG tasks.

[4] See description of the medical team in Appendix A.3.

| Disease Codes of MEDEVAL | | | |
| --- | --- | --- | --- |
| Enlarged Cardiomediastinum | Cardiomegaly | Lung Opacity | Lung Lesion |
| Edema | Consolidation | Atelectasis | Pneumothorax |
| Pleural Effusion | Pleural Other | Support Devices | Pneumonia |
| Dislocation | Osteonecrosis | Fracture | Gout |
| Metatarsus Primus Varus | Gas | Swelling | Psoriasis |
| Enthesopathy | Hammer Toe | Osteomyelitis | Mass |
| Arthritis | Pes Planus | Rheumatoid | Cppd |
| Hardware | Erosion | Pes Cavus | Coalition |
| Subluxation | Fracture | Nodule | Rupture |
| Hallux Valgus | Pneumonia | Arthritis | No Finding |

Table 1: Report disease codes covered in MEDEVAL.

## 3.1 Input Data Composition

**Sentence-Level Corpora** The sentence-level corpora used in this study are sourced from two well-constructed datasets: the sentence-level OpenI-annotated dataset (Demner-Fushman et al., 2016), which consists of sentences from chest studies, and the VA-annotated dataset (He et al., 2023b), which includes sentences about different body parts examined by different modalities. These datasets have undergone de-identification, completion of missing terms and uniqueness checks. More details about the data preprocessing is given in Appendix A. We use the officially released versions of the OpenI-annotated and VA-annotated datasets. In addition, we provide new annotations for sentence-level tasks on these data sources.

**Report-Level corpora** We collect the raw radiology reports from two distinct sources: (1) text corpus from MIMIC-CXR, which comprises records related to human chests (Johnson et al., 2019), (2) text corpus from the databases of a nationwide government healthcare system. We randomly collect data points about different body parts and exam modalities, resulting in multiple domains under different data distributions. The distribution of the domain is illustrated in Figure 1. The collected data are processed with automatic de-identification, followed by a thorough human inspection to verify that no private information about patients or doctors is disclosed or hinted at in the text. We also employ an offline paraphrasing tool (Damodaran, 2021) to revise the text data collected from the second source. The paraphrasing is followed by another human inspection to filter out any unqualified records where the rewriting deviates significantly from the original report. The resulting data set can be considered "synthesized" and containing no privacy information but retaining realistic clinical conditions as the source data.

For each evaluation task, we split the data in a ratio of 7:1:2 for train/validate/test.

## 3.2 Expert Labels and Evaluation Tasks

## 3.3 Sentence-level Labels

**NLU Tasks** Identifying sentences with certain diagnostic properties is a practical use case in a real-world healthcare system. For example, identifying if a report sentence implies an abnormal finding about the patient or not. To test if language models can capture the medical semantics of single sentences, we first include abnormal sentence identification into our evaluation pool. We use the sentence-level corpora and the associated abnormality labels to classify abnormal sentences.

Ambiguous sentences appear in radiology reports mainly due to the use of medical jargon whose meaning is different from daily usage, contradictory findings within the same sentence, or grammatical errors that mislead interpretation (He et al., 2023b). Accurate identification of such sentences is crucial, as they impede patients' comprehension of diagnostic decisions, leading to potential treatment delays and irreparable consequences. To the best of our knowledge, as a novel task proposed recently, current LMs may not readily include such a task into its pre-training stage. Therefore, evaluation of this task allows us to investigate how language models perform when the tasks are unfamiliar. We leverage the report sentences and their associated ambiguous labels, and our medical team re-examined and re-annotated the labels for ambiguous sentences.

**NLG Task** Expanding beyond the previous ambiguous sentence identification, we include the task of sentence disambiguation as a sentence-level generation task. Proposed in He et al. (2023b), sentence disambiguation aims to rewrite an ambiguous sentence in a way that its diagnostic findings are more explicitly expressed while at the same time, the original content of the report sentence is faithfully maintained. This requires rewritten sentences to avoid the change of the original pathological findings or introducing new findings. Similar to ambiguous sentence identification, disambiguated rewriting presents a challenging generation task, not only because both the data and task formulation are not likely to be covered in the pre-training stage of existing language models, but also because there are two objectives that need to be optimized at the same time. In this task, based on the ambiguous sentences and their associated diagnostic labels, our medical team manually created the dis-

ambiguated rewritings as the ground truth.

### 3.4 Document-level Labels

**NLU Task** To access if language models can capture the key findings of a radiology report, we consider Report Codes Prediction as an evaluation task. This task involves categorizing reports into specific diagnostic codes based on the mentioned pathological findings. Therefore, different from sentence-level abnormality identification, this task requires a multi-label multi-class classification. Our medical team manually labels the medical codes of each report. Detailed information regarding the codes is provided in Table 1. More details about the expert-labeling procedure are provided in Appendix A.

**NLG Task** Automatic medical summarization plays a crucial role in healthcare literature, by providing concise summaries, it saves time and manual effort for medical professionals when assessing the effectiveness of medical interventions. In our evaluation, we include report summarization as a task to assess the generation capability of language models. The *impression* section in each report serves as a summary that captures the supportive evidence for clinical decisions. To ensure data quality, we conduct a manual inspection of all collected <report, impression> pairs, filtering out any pairs where the impression does not align with the corresponding report. It is worth noting that the curated parallel data of reports and summaries provide valuable support for future work in related fields.

### 3.5 Evaluated Language Models

We evaluate two categories of language models with MEDEVAL[5]: (1) domain-adapted pre-trained language models (Adapted PLMs), which are trainable models adapted on certain domain data, and (2) general-purpose large language models (Prompted LLMs) which are used by zero/few-shot prompting.

### 3.5.1 Domain-adapted PLMs

Recent literature found it is effective to adapt pre-trained language models to certain narrow domains such as biomedical text by a continued training step on domain-specific data (Gururangan et al., 2020a), following which we take a pre-trained (or generally adapted) language model, and test it on the MEDEVAL test set. We also fine-tuned the models

---

[5]The results presented are based on models evaluated as of the time of paper acceptance. We've added more results (e.g., on GPT4, LLaMa2, etc.) and will continue to include the newest ones in the Appendix E.

from this category to customize it to fit the tasks of MEDEVAL, with their corresponding training data. For NLU tasks at both levels, we follow the evaluation setting of Yan et al. (2022) and investigate how: **BERTbase** (Devlin et al., 2018), **RadBERT** (Yan et al., 2022), **BioBERT** (Lee et al., 2020), **clinicalBERT** (Huang et al., 2019), **BlueBERT** (Peng et al., 2019), and **BioMed-ReBERTa-base** (Gururangan et al., 2020b) perform on MEDEVAL. More details about those models are included in Section 2.

For the sentence-level NLG task, we follow the the setting of He et al. (2023b) by evaluating: (1) **style transformer** (Dai et al., 2019) which transfers the original sentence into a less ambiguous style, (2) **PPLM** (Dathathri et al., 2020) which adds perturbation to LM to move the (re-)generation towards a less ambiguous direction, (3) **DEPEN** (He et al., 2021a) which is built upon PPLM and only re-generates ambiguous tokens detected before, and (4) **MedDEPEN** (He et al., 2023b), a biomedical-adapted DEPEN by introducing contrastive pre-training. Each work has included a transformer-based language model. We refer the reader to the original papers for more details.

For the document-level NLG task, we follow the setting of Yan et al. (2022) and customize previously adapted BERT-based models used before for the summarization task.

### 3.5.2 Prompted LLMs

We include the following general-purpose large language models to test their generalization in the healthcare domain: (1) **GPT3**: GPT-style large language models with 175B parameters (Brown et al., 2020). We use davinci-003. (2) **Chat-GPT**[6]: GPT-style large language model trained with Reinforcement Learning from Human Feedback (RLHF). We use GPT3.5-turbo. (3) **Vicuna-7B** (Chiang et al., 2023): The finetuned version of LLaMa-7B (Touvron et al., 2023a) with 70K user-shared ChatGPT conversations, which is capable of generating more detailed and well-structured answers. (4) **BioMedLM**[7]: a 2.7B GPT-style language model trained exclusively on biomedical abstracts and papers from The Pile(Gao et al., 2020).

We prompt those LLMs under zero/few-shot settings, where we randomly select the examples from the training set of each task to compose prompts 5

---

[6]https://openai.com/blog/chatgpt
[7]https://crfm.stanford.edu/2022/12/15/biomedlm.html

| Models | Chest | | Miscellaneous Domains | |
|---|---|---|---|---|
| | Abnormality ↑ | Ambiguity ↑ | Abnormality ↑ | Ambiguity ↑ |
| **Adapted PLMs with Fine-Tuning** | | | | |
| BERT | 0.9791 | **0.9893** | 0.9607 | 0.9749 |
| RadBERT | 0.9794 | 0.9869 | 0.9640 | **0.9813** |
| BioBERT | 0.9791 | 0.9862 | 0.9614 | 0.9743 |
| ClinicalBERT | **0.9809** | 0.9874 | 0.9588 | 0.9736 |
| BlueBERT | 0.9803 | 0.9867 | 0.9601 | 0.9775 |
| BioMed-ReBERTa | 0.9569 | 0.9758 | **0.9776** | 0.9788 |
| **LLMs Prompted by Zero/Few Shot** | | | | |
| zero-shot ChatGPT | 0.9277 | 0.6584 | 0.8880 | 0.5206 |
| few-shot ChatGPT | 0.9498 | 0.5831 | 0.9099 | 0.5354 |
| zero-shot GPT-3 | 0.8762 | 0.8742 | 0.8243 | 0.6448 |
| few-shot GPT-3 | 0.9215 | 0.8320 | 0.9054 | 0.6371 |
| zero-shot Vicuna-7B | 0.6987 | 0.2130 | 0.7261 | 0.3739 |
| few-shot Vicuna-7B | 0.8071 | 0.0785 | 0.8166 | 0.2844 |
| zero-shot BioMed LM | 0.6679 | 0.3485 | 0.6273 | 0.3726 |
| few-shot BioMedLM | 0.7905 | 0.6804 | 0.7638 | 0.6804 |

Table 2: Evaluation (accuracy) over two categories of PLMs on abnormality identification and ambiguity identification tasks (sentence-level NLU). **Bold**: the highest performance. Underlined: the lowest.

| Model | | Chest | | | Miscellaneous Domains | | |
|---|---|---|---|---|---|---|---|
| | | Disambiguation $\Delta Acc_{am}$ ↑ | Content Distortion $\Delta Acc_{ab}$ ↓ | BLEU4 ↑ | Disambiguation $\Delta Acc_{am}$ ↑ | Content Distortion $\Delta Acc_{ab}$ ↓ | BLEU4 ↑ |
| **Adapted PLMs with Fine-Tuning** | Style Transfer | 0.5010 | 0.0510 | 27.92 | 0.3110 | 0.2350 | 31.17 |
| | PPLM | 0.3860 | 0.1150 | 57.88 | 0.2700 | 0.1460 | 60.14 |
| | DEPEN | 0.5000 | 0.0520 | 60.48 | 0.3530 | 0.0470 | 67.86 |
| | MedDEPEN | 0.4960 | 0.0320 | 57.88 | 0.4810 | **0.0090** | 68.88 |
| **LLMs Prompted by Zero/Few Shots** | zero-shot ChatGPT | 0.6337 | **-0.0297** | 60.73 | 0.6539 | 0.1483 | 60.64 |
| | few-shot ChatGPT | 0.5875 | 0.0000 | 68.92 | 0.6370 | 0.0815 | 67.98 |
| | zero-shot GPT-3 | **0.6799** | -0.0132 | 61.78 | **0.8022** | 0.1528 | 61.05 |
| | few-shot GPT-3 | 0.6139 | 0.0000 | **76.33** | 0.7146 | 0.0607 | **77.09** |
| | zero-shot Vicuna-7B | 0.6230 | 0.0693 | 66.65 | 0.6771 | 0.3653 | 64.64 |
| | few-shot Vicuna-7B | 0.5311 | 0.1914 | 62.55 | 0.4811 | 0.2739 | 63.72 |
| | zero-shot BioMed LM | 0.2211 | 0.0066 | 23.40 | 0.1528 | 0.1416 | 24.11 |
| | few-shot BioMed LM | 0.1386 | -0.0262 | 23.30 | 0.3933 | 0.3640 | 23.48 |

Table 3: Evaluation on disambiguated rewriting Tasks (sentence-level NLG). We report the disambiguation score, content distortion score (where smaller content distortion indicates higher fidelity), and BLEU4 score. **Bold**: the best performance. Underlined: the worst.

times. We report the test results with the prompts which obtain optimal results on the validation set. See Appendix C for more details.

## 3.6 Evaluation Metrics

For NLU tasks, we report classification metrics including accuracy and F1 scores. For NLG tasks, we report BLEU and ROUGE scores with respect to the ground truths labeled by our medical team. For sentence-level generation tasks (i.e., rewriting), to evaluate the objective of disambiguation, we follow the setting of He et al. (2023b) to report accuracy decrements of the ambiguity classifier ($\Delta Acc_{am}$) as the disambiguation metric. To evaluate the rewriting fidelity, we report the content distortion score, which is defined as the decrement of the accuracy from an abnormality classifier

($\Delta Acc_{ab}$). Therefore, higher distortion indicates a lower content fidelity.

## 4 Results and Discussion

In this section, We first present the results for sentence-level NLU tasks (Ambiguity Identification and Abnormality Identification) in Table 2, then sentence-level NLG task (Disambiguated Rewriting) in Table 3, finally document-level NLU (Code Prediction) and NLG (Report Summarization) tasks in Table 4 and Table 5.

**The Effectiveness of Instruction Tuning** While BioMed LM is the first large language model customized for the biomedical domain, we observe that it does not outperform adapted PLMs and most prompted LLMs in the majority of tasks. Particu-

| Model | Chest | | Foot | | Ankle | |
|---|---|---|---|---|---|---|
| | avg Accuracy ↑ | avg EMR ↑ | avg Accuracy | avg EMR ↑ | avg Accuracy ↑ | avg EMR ↑ |
| **Adapted PLMs with Fine-Tuning** | | | | | | |
| BERT | 0.8779 | 0.2263 | **0.9754** | 0.5635 | 0.9787 | 0.6141 |
| RadBERT | 0.8785 | 0.1941 | 0.9710 | 0.4910 | 0.9773 | 0.5710 |
| BioBERT | 0.8782 | **0.2400** | 0.9750 | 0.5617 | **0.9801** | **0.6266** |
| ClinicalBERT | 0.8780 | 0.2341 | 0.9731 | 0.5372 | 0.9798 | 0.6224 |
| BlueBERT | **0.8843** | 0.2380 | 0.9703 | **0.5939** | 0.9761 | 0.5752 |
| BioMed-ReBERTa | 0.8579 | 0.1415 | 0.9692 | 0.4631 | 0.9752 | 0.5522 |
| **LLMs Prompted by Zero/Few Shots** | | | | | | |
| zero-shot ChatGPT | 0.5272 | 0.1024 | 0.9621 | 0.4449 | 0.9660 | 0.4491 |
| few-shot ChatGPT | 0.6485 | 0.1951 | 0.9621 | 0.4186 | 0.9690 | 0.4875 |
| zero-shot GPT-3 | 0.2744 | 0.1424 | 0.9621 | 0.4449 | 0.1887 | 0.6273 |
| few-shot GPT-3 | 0.8160 | 0.1805 | 0.9617 | 0.4186 | 0.9691 | 0.4908 |
| zero-shot Vicuna-7B | 0.8216 | 0.0672 | 0.9617 | 0.4186 | 0.9691 | 0.4908 |
| few-shot Vicuna-7B | 0.8228 | 0.0782 | 0.5156 | 0.1041 | 0.9122 | 0.4153 |
| few-shot BioMed LM | 0.8320 | 0.0689 | 0.9667 | 0.4664 | 0.9719 | 0.4980 |

Table 4: Evaluation on report codes prediction Task (Document-level NLU). We report the average accuracy over all classes of diseases and the exact match rate (EMR) between predictions and labels. **Bold**: the highest performance. Underlined: the lowest.

| Model | Miscellaneous Domains | | | | |
|---|---|---|---|---|---|
| | ROUGE-1 ↑ | ROUGE-2 ↑ | ROUGE-L↑ | Sum ↑ | BLEU4↑ |
| **Adapted PLMs with Fine-Tuning** | | | | | |
| BERT | 20.48 | 7.46 | 18.57 | 46.52 | 30.28 |
| RadBERT | 20.96 | 7.63 | 18.90 | 47.50 | 30.77 |
| BioBERT | 20.79 | 7.62 | 18.78 | 47.19 | 30.49 |
| ClinicalBERT | 21.22 | 7.85 | 19.18 | 48.26 | 30.81 |
| BlueBERT | 20.83 | 7.78 | 18.90 | 47.51 | 30.93 |
| BioMed-ReBERTa | 21.19 | 7.85 | 19.14 | 48.18 | 30.88 |
| **LLMs Prompted by Zero/Few Shots** | | | | | |
| zero-shot ChatGPT | 24.64 | 7.97 | 22.05 | 54.66 | 32.30 |
| few-shot ChatGPT | **24.96** | 8.43 | **22.23** | 55.62 | **35.43** |
| zero-shot GPT-3 | 24.06 | 8.67 | 21.52 | 54.24 | 24.43 |
| few-shot GPT-3 | 24.73 | **9.14** | 22.16 | **56.03** | 34.72 |
| zero-shot Vicuna-7B | 20.93 | 6.96 | 18.71 | 46.60 | 20.94 |
| few-shot Vicuna-7B | 21.42 | 7.26 | 19.22 | 47.90 | 22.00 |
| zero-shot BioMed LM | 17.70 | 5.11 | 16.49 | 39.30 | 15.43 |
| few-shot BioMed LM | 12.15 | 3.50 | 11.22 | 26.87 | 20.27 |

Table 5: Evaluation on report summarization task (Document-level NLG). We report the Rouge scores and BLEU4 scores. **Bold**: the highest performance. Underlined: the lowest.

larly, BioMed LM has been found to be the weakest performer in tasks such as sentence identification, disambiguated rewriting, and report summarization. We would like to highlight that, unlike other prompted LLMs such as ChatGPT, GPT-3, and Vicuna, BioMed LM lacks an Instruction Tuning step in its model training. This omission significantly impacts BioMed LM's ability to generate replies following the instructions from the given options. In zero-shot NLU tasks, only 40% of the test cases receive appropriate responses at the sentence level and the qualified rate drops to less than 1% at the document level (so we did not report the results in Table 4). In few-shot report codes prediction, the document-based prompts often exceed BioMed LM's maximum threshold of 1024 tokens, resulting in query errors. In generation tasks, BioMed LM keeps returning irrelevant text. Our manual inspection reveals that the outputs rarely adhere to the given instructions in prompts or address the queries. This is further supported by the remarkably low BLEU or ROUGE scores in Table 3 and Table 5. We provide more discussions in Appendix D.1. These findings underscore the significance of Instruction Tuning and establish it as a crucial step when adapting prompted LLMs for specialized applications like healthcare decision-making.

In the remainder of this section, we focus on addressing more intriguing questions based on average performance across a range of baselines (e.g., the average accuracy of adapted PLMs versus prompted LLMs), where we exclude BioMed LM from further consideration.

**Discussion on Task Type and Granularity** In this section, we aim to determine the proficiency of language models at different levels and tasks. To achieve this, we begin by calculating the average accuracy scores of all adapted PLM baselines and

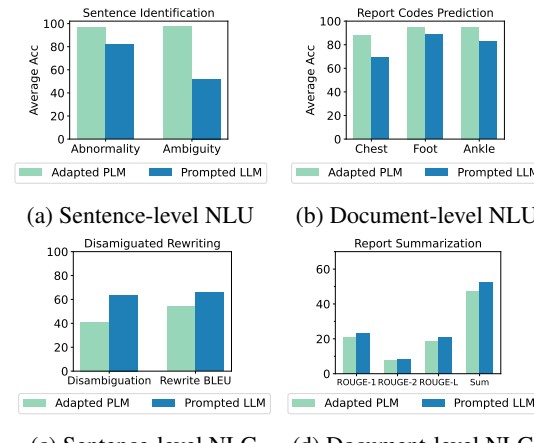

(a) Sentence-level NLU     (b) Document-level NLU

(c) Sentence-level NLG     (d) Document-level NLG

Figure 3: Average performance of adapted PLM and prompted LLM on different tasks and at different levels.

| Model | Abnormality ↑ | | Ambiguity ↑ | |
|---|---|---|---|---|
| | Chest | Miscellaneous | Chest | Miscellaneous |
| Adapted PLM | 0.9758 | 0.9526 | 0.9836 | 0.9621 |
| Prompted LLM | 0.8635 | 0.8451 | 0.5399 | 0.4893 |

Table 6: Average accuracy of adapted PLMs and prompted LLMs in NLU over different domains

prompted LLM baselines in sentence identification tasks. Similarly, we compute the average accuracy of adapted PLM and prompted LLM baselines in a document-level code classification task.

First, examining the results presented in Figure 3, we observe that both adapted PLMs and prompted LLMs perform relatively similarly across different data levels. However, it becomes apparent that adapted PLMs outperform prompted LLMs in NLU tasks, no matter whether it's on the sentence or document level. This suggests that fine-tuning provides a more effective means of injecting specific knowledge about narrow domains or tasks. On the other hand, consistently superior performance of prompted LLMs compared to adapted PLMs is observed in generation tasks, at both the sentence and document levels. This can be attributed to multiple advantages of large-scale pre-training such as a larger model size or the benefits HFRL in the LLMs we utilized, such as ChatGPT. These models demonstrate a capability to generate language that is more akin to human-like expressions, thereby achieving better generation scores. These imply that fine-tuning PLM models can be a viable choice for NLU tasks, while prompting-based LLMs may be more suitable when healthcare professionals require an AI writer to help their work.

| Model Family | # shot | NLU (Accuracy↑) | | NLG (BLEU↑) | |
|---|---|---|---|---|---|
| | | Individual | Average | Individual | Average |
| ChatGPT | 0-shot | 0.78 | 0.79 | 51.22 | 47.15 |
| | Few-shot | 0.79 | | 43.08 | |
| GPT-3 | 0-shot | 0.66 | 0.76 | 49.08 | 55.90 |
| | Few-shot | 0.86 | | 62.71 | |
| Vicuna-7B | 0-shot | 0.71 | 0.73 | 50.74 | 50.08 |
| | Few-shot | 0.75 | | 49.42 | |
| Average | 0-shot | 0.72 | | 50.35 | |
| | Few-shot | 0.80 | | 51.74 | |

Table 7: Average accuracy and BLEU of various LM families with zero/few shots.

**Common v.s. Rare Domains** In Table 6, we explore the impact of the domain on language models in the healthcare field. We compute the average accuracy of adapted PLMs and prompted LLMs in abnormality identification v.s. ambiguity identification. We consistently observe higher performance from both adapted PLMs and prompted LLMs when working with data from the chest domain compared to miscellaneous domains. This superior performance can be attributed to the similarity between the chest data we tested and the pre-training data of the language models – chest-related healthcare text is widely available in the public domain and can be included in the training corpus of PLMs. Similarly, LMs are expected to excel in abnormality identification tasks, which are a common research topic in current literature.

The most challenging scenario arises when both the data and task are unseen, specifically in the case of ambiguous identification within the miscellaneous domain. In such situations, there are limited or no examples available in the public domain. Therefore, querying language models with (zero) few-shot learning proves to be less effective.

**Family of LLMs and Few Shot Learning** In this analysis, we examine the behavior of different language models (LLMs) with varying numbers of shots across different tasks. We calculate the average accuracy of ChatGPT, GPT3, and Vicuna-7B in NLU tasks and the average BLEU scores in NLG tasks. Additionally, we consider the average performance achieved in zero-shot or few-shot settings (Table 7). From the table, it is evident that in most cases, providing additional examples assists LMs in making predictions for NLU tasks. However, in NLG tasks, no consistent trend is observed, indicating the need for further research to discover optimal prompts. We do not observe a clear advantage of any specific LLM family over others, suggesting

that the choice of the optimal LLM family for a given task may vary on a case-by-case basis.

## 5 Conclusion

We introduce MEDEVAL, a multi-task, multi-level, and multi-domain medical benchmark designed to serve as a comprehensive testbed for advanced language models. Through extensive evaluation experiments, we thoroughly analyze the capabilities and limitations of current LLMs in tackling various medical tasks, such as the effectiveness of instruction tuning and the performance disparities between adapted and prompted LMs in NLU and NLG tasks. Our findings provide valuable insights and serve as a handbook for future research in utilizing LLMs to enhance healthcare practices.

## 6 Limitations

In our efforts to provide a comprehensive testbed for current advanced language models, we have included multiple tasks. However, we acknowledge that there may be other tasks of interest that could have been analyzed, such as medical named entity recognition, multi-document report summarization, etc. We plan to expand the range of test tasks in future iterations of the MEDEVAL benchmark. We'd like to note that due to computing constraints, we were unable to evaluate some large language models such as Vicuna-60B or OPT-175B (Zhang et al., 2022). Our evaluation was focused on popular large language models with reasonably large sizes. In future work, we consider addressing this limitation by incorporating these larger language models into our testbed.

## 7 Ethics Statement

Our data underwent a rigorous de-identification process and were carefully reviewed by human evaluators following strict anonymization criteria. Moreover, the collection of data from real-world healthcare systems has received the necessary IRB approval (DC VAMC protocol 1736644, VASDHS IRB protocol 200086), ensuring compliance with ethical standards. To further ensure ethical usage, before inputting the data into large language models, including commercial ones like ChatGPT, we conducted data synthesis and subjected it to additional human inspection. These steps were taken to address any potential ethical concerns associated with the data.

It is important to highlight that the responsible and safe usage of biomedical data is a critical requirement in AI for healthcare, especially in the use case of large language models which are noticed to suffer from different kinds of potential harms (Leino et al., 2019; Lloyd, 2018; He et al., 2021b; Xu et al., 2022; He et al., 2022) and weaknesses (Ribeiro et al., 2020; Stuart-Ulin, 2018; He et al., 2023a). Therefore, we strongly recommend that our benchmark be used in conjunction with expert auditing to ensure the highest level of safety in real-world applications.

## Acknowledgments

The authors would like to thank NVIDIA corporation for the award of two RTX A6000 GPUs to CNH through the Applied Research Accelerator Program.

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

## A Data Preparation

### A.1 Preprocess of sentence-level corpora

**OpenI** In the original OpenI release, many non-sensitive terms were incorrectly masked as "xxxx" by the de-identification software described in (Demner-Fushman et al., 2016). Our medical team manully fills in the missing information based on the context of the reports and additional information associated with it.

### A.2 Preprocess of the document-level corpra

**Manual De-identification Criteria** We hire human reviewers to manually inspect the reports after the automatic de-identification tools. According to our criteria, we will discard a datapoint if it contains

- real names of the patient, or the healthcare professions,

- home address, working address, or locations of the patient or healthcare professionals.

- contact information (e.g., phone number) about the patient, or healthcare professionals.

In the second round of human inspection about de-identification, 99.8% of the data are well de-identified in the automatic stage, and 0.2% of the data are discarded.

**Diseases Code Preparation** Before experts examine the disease codes of a report, we first group data by their resources. For reports sourced from MIMIC-CXR, we employ CheXpert (Irvin et al., 2019), a rule-based automatic labeler to generate pseudo-labels for the diagnostic codes. Each label has three options: positive, negative, and unknown. In the case of reports from the second source, we customized a rule-based automatic labeler called `pyConText NLP`[8], which generates pseudo-codes according to the keywords of each domain. In the last step, our medical team manually reviews, correct the codes where there is a conflict (e.g., positive "no finding" appears with certain positive diseases), and writes down the correct codes for each report.

### A.3 Medical Team

Our medical expert team consists of 4 members, including 2 senior board-certified radiologists with

more than 15 years of experience in healthcare and a doctor who has more than 10 years of experience serving as a PI of medical research. We follow standard labeling practices, involving multiple rounds of iterative review by different experts until Cohen's kappa coefficient reaches 0.85. Any remaining disagreements are collectively resolved.

## B Implementation details

**Models** All adapted transformers are implemented based on the HuggingFace[9] libraries. All prompted LLMs are implemented with its original release in their official webpage or GitHub.

The model sizes are listed in the Section 2.

**Experiment Hyperparameters** The configurations of querying LLMs are listed as follows:

|  | GPT-3 | ChatGPT |
|---|---|---|
| Engine | text-davinci-003 | gpt-3.5-turbo |
| Temperature | 0 | 0 |
| Max Tokens | 200 | 200 |
| Return N | 1 | 1 |
| Top P | 1 | 1 |
| Frequency Penalty | 0 | 0 |
| Presence Penalty | 0 | 0 |

Table 8: Configuration of ChatGPT and GPT-3

The configurations for fine-tuning adapted languages models are: learning rate=0.0001, weight decay=0, optimizer=Adam, training epoch=10, batch size=64, max length=256, All codes are implemented with Python3.8 and PyTorch1.7.1 with CUDA10.1. operated on Ubuntu (16.04.7 LTS) server with 2 NVIDIA GeForce GTS A6000 GPUs. Each has memory of 49GB.

## C Prompts used for Querying LLM

### C.1 Number of examples in few-shot settings

In our study, we maintain a balanced and unbiased approach by setting the number of examples equal to the number of classes when prompting the language models (especially in NLU tasks). Additionally, we explore alternative numbers of examples, such as 1, 3, 5, 7, or 9. The NLU experiment results presented in our paper utilize a 2-shot approach, while the NLG results employ a 3-shot approach.

---

[8] https://github.com/chapmanbe/pyConTextNLP

[9] https://huggingface.co/models

## C.2 Templates Used in Prompting

For zero-shot settings we design prompts using the following template:

We are here to address a new task, [TASK NAME]. Given a [LEVEL NAME], written by a radiologist, please [TASK CONTENT]. Now here is a new [LEVEL], tell me [TASK CONTENT], without saying anything else. Input: {}, Label:

For few-shot settings we insert examples extracted from the training set of each task into the prompts by:

We are here to address a new task, [TASK NAME]. Given a [LEVEL NAME], written by a radiologist, please tell [TASK CONTENT]. Here are [N] examples. [EXAMPLE 1]...[EXAMPLE N] Now here is a new [LEVEL], tell me [TASK CONTENT], without saying anything else. Input: {}, Label:

The options for the placeholder are:

- TASK: {*abnormal identification, ambiguous identification, rewrite an ambiguous sentence to be less ambiguous, summarization* }

- LEVEL NMAE: {*sentence, report*}.

- TASK CONTENT:{*Tell if the sentence indicates abnormal findings or not, Tell if it is ambiguous, a sentence is defined to be ambiguous because of (1) medical jargon with meanings different from everyday general usage, such as unremarkable; (2) contradictory findings in the same sentence; (3) misleading grammatical errors such as no period between full sentences, , Observed pathology findings, The summary* }

- N: {1, 2, 3, 4, 5, 7, 9}

In the following sections, we provide some examples of the prompts we use to query LLM.

## C.3 Sentence-Level Tasks

For sentence-level tasks, there are the following different types of prompts for NLU tasks:

**Zero-shot Classification for Abnormality** We are here to address a new task, abnormal identification. Given a sentence from a radiology report, written by a radiologist, please tell if this sentence indicates abnormal findings or not.
Now here is a new sentence, tell me if it is normal or abnormal, without saying anything else. Sentence: {} Label:

**Two-shot classification for Abnormality:** We are here to address a new task, abnormal identification. Given a sentence from a radiology report, written by a radiologist, please tell if this sentence indicates abnormal findings or not. Here are 2 examples.
Sentence: there is likely left basilar opacity. Label: abnormal.
Sentence: unchanged exam without acute abnormality. Label: normal.
Now here is a new sentence, tell me if it is normal or abnormal, without saying anything else.
Sentence: {} Label:

**Zero-shot Classification for Ambiguity:** Here is a task to classify ambiguous sentences. Given a sentence in a radiology report, written by a radiologist, please tell if it is ambiguous. A sentence is defined to be ambiguous because of (1) medical jargon with meanings different from everyday general usage, such as unremarkable; (2) contradictory findings in the same sentence; (3) misleading grammatical errors such as no period between full sentences.
Now given a new sentence, answer me with "ambiguous" or "unambiguous", without saying anything else. Sentence: {} Label:

**Two-shot Classification for Ambiguity:** Here is a task to classify ambiguous sentences. Given a sentence in a radiology report, written by a radiologist, please tell if it is ambiguous. A sentence is defined to be ambiguous because of (1) medical jargon with meanings different from everyday general usage, such as unremarkable; (2) contradictory findings in the same sentence; (3) misleading grammatical errors such as no period between full sentences. Here are 2 examples.
Sentence: lungs are unremarkable. Label: ambiguous.
Sentence: unchanged chronic appearance of the left lung. Label: unambiguous.
Now given a new sentence, answer me with

"ambiguous" or "unambiguous", without saying anything else.
`Sentence: {} Label:`

Similarly, we will add more examples for the prompts built for more than two-shot situations. For NLG tasks, we have the following prompts:

**Zero-shot Rewriting tasks** `Here is a task to rewrite ambiguous sentences to be less ambiguous. Given a sentence in a radiology report, written by a radiologist, please rewrite it to be more explicit about the diagnostic decision reflected in the sentence, however, maintain the main meaning of the original sentence. A sentence is defined to be ambiguous because of (1) medical jargon with meanings different from everyday general usage, such as unremarkable; (2) contradictory findings in the same sentence; (3) misleading grammatical errors such as no period between full sentences. Now given a new sentence, answer me with its rewrite, without saying anything else. Sentence: {}. Rewrite:`

**Three-shot Rewriting tasks** `Here is a task to rewrite ambiguous sentences to be less ambiguous. Given a sentence in a radiology report, written by a radiologist, please rewrite it to be more explicit about the diagnostic decision reflected in the sentence, however, maintain the main meaning of the original sentence. A sentence is defined to be ambiguous because of (1) medical jargon with meanings different from everyday general usage, such as unremarkable; (2) contradictory findings in the same sentence; (3) misleading grammatical errors such as no period between full sentences. Here are three examples.`
`Sentence: Lungs are unremarkable. Diagnostic: Normal. Rewrite: Lungs are normal.`
`Sentence: The lung volumes are low normal. Diagnostic: Normal. Rewrite: The lung volumes are in the lower range of normal limit.`
`Sentence: Cardiomegaly and hiatal hernia without an acute abnormality`
`identifie. Diagnostic: Abnormal. Rewrite: Cardiomegaly and hiatal hernia . Without an acute abnormality identified. Now given a new sentence, answer me with its rewrite, without saying anything else. Sentence: {}. Rewrite:`

## C.4 Document-level Tasks

For NLU tasks, we use the following prompts:

**Zero-shot Summarization Tasks** `Here is a summarization task. Given a radiology report written by a radiologist, please write a summary of the report. Answer me with its summary only, without saying anything else. Report: {}. Summary:`

**Three-shot Summarization Tasks** `Here is a summarization task. Given a radiology report written by a radiologist, please write a summary of the report. Here are 5 examples.`
`Report: clinical history 62-year-old male with bilateral bunions. please perform weighted views. a lateral weight bearing view of bilateral feet as well as a lateral non-weight bearing view of bilateral feet are studied. three views of the right foot show no fracture dislocation foreign body pathologic calcification or soft tissue swelling. the joint spaces are not noticeable. there is a deformity of the hallux valgus. the angle of pitch is within normal limits. three views of the left foot show no fracture dislocation foreign body pathologic calcification or soft tissue swelling. the joint spaces are not noticeable. hallux valgus minor. the angle of pitch is within normal limits. Summary: 1 bilateral hallux valgus deformities. 2 no acute osseous abnormalities. ssn7312ptc1job no. 1157.`
`Report: there has been no significant change in the patient's condition since the patient's exam which was earlier in the day. the heart size is normal and the lungs are free of disease. on the left base is again noted a small granulom. Summary: for a active disease in the chest there is no evidence.`
`Report: the cardiovascular-mediastinal silhouette is normal. it's not unusual`

```
for pulmonary vessels. the bones appear
to be intact.  Summary:  chest x-rays
within normal ranges. no change in date
2010-06-28.
Now given a new report, answer me with
its summary only, without saying anything
else. Report: {}. Summary:
```

## D  More Discussions

### D.1  Case Studies of BioMed LM

BioMed LM, a GPT-style LLM trained on PubMed, lacks instruction fine-tuning in its training process. As a result, the model's outputs often lack a unified format, making it challenging to conduct follow-up statistical evaluations. To assess the model's performance, we introduce the concept of a qualified rate, which represents the proportion of test cases where the model provides a relevant prediction for the given task, such as identifying abnormalities by including only one of the "normal" or "abnormal" in the response. We report the qualified rate here:

|  | Sentence | Report |
| --- | --- | --- |
| Zero-shot NLU | 40% | 1% |
| Few-shot NLU | 92% | 85% |

Table 9: Qualified Rate of BioMed LM on NLU tasks

In zero-shot settings, BioMed LM struggles to adhere to the instructions in the prompt, generating outputs that are freestyle and not aligned with the expected format. In a few-shot setting, particularly in document-level tasks, the length of the prompts exceeds the maximum input capacity of BioMed LM (1024 tokens). To address this issue, we employ two solutions: (1) chunking the input into 1024-token segments and (2) discarding test cases that exceed the maximum length. However, both solutions have drawbacks. The chunked un-completed input approach leads to more "freestyle" outputs, resulting in a lower qualified rate. On the other hand, discarding examples introduces a high variance in the statistics. We show some examples of the "freestyle" outputs at the end of this section.

Considering these challenges, we treat BioMed LM as an exception and exclude it when analyzing results across different categories of LMs. Nonetheless, we emphasize the importance of further investigation into biomedical LLMs for future research.

**Example Output of BioMed LM**  We show some examples of unqualified "free-style" outputs

from BioMed LM in the following:

**Sentence Level, NLU-Zero-shot**  Note that we expect the model to return a single word from "normal", "abnormal", "ambiguous", or "unambiguous". However, BioMed LM returns the following output:

```
Prompt: Given a sentence from a radiology
report,  written  by  a  radiologist,
please  tell  if  the  following  sentence
indicates "normal" or "abnormal" findings.
Sentence: median sternotomy wires appear
intact. Label:
Returned Text: median  sternotomy  wires
appear intact.
```

**Sentence Level, NLU-Two-shot**  In this case, we expect the model to return a single word "normal", "abnormal", "ambiguous", "unambiguous". However, the output from BioMed is:

```
Prompt: Here  is  a  task  to  classify
ambiguous  sentences.  Given  a  sentence
in  a  radiology  report,  written  by  a
radiologist,  please  tell  if  it  is
ambiguous.  A  sentence  is  defined  to  be
ambiguous  because  of  (1)  medical  jargon
with  meanings  different  from  everyday
general  usage,  such  as  unremarkable;
(2)  contradictory  findings  in  the  same
sentence;  (3)  misleading  grammatical
errors  such  as  no  period  between  full
sentences. Here are 2 examples.
Sentence: lungs are unremarkable. Label:
ambiguous.
Sentence: unchanged chronic appearance of
the left lung. Label: unambiguous.
Now given a new sentence, answer me with
"ambiguous" or "unambiguous".
Sentence: findings:  there  is  a  cystic,
approximately  3.  6  x  2.  6  x
6.  6  cm,  mass  with  its  epicenter
in  the  left  submandibular  region,  with
extension  into  the  left  submental  space,
left  sublingual  space,  left  base  of
tongue,  and  superiorly  into  the  left
peripharyngeal space. Label:
Returned Text: cystic mass.
Sentence:      there   is   a   cystic,
approximately  3.  6  x  2.  6  x  6.  6
cm, mass with its epicenter in the left
submandibular  region,   with  extension
```

into the

**Sentence Level, NLG-Zero-shot**   Here we show an unqualified example in the ambiguous rewriting task, where the output does not follow the instruction of "without saying anything else".

Prompt: `Here is a task to rewrite ambiguous sentences to be less ambiguous. Given a sentence in a radiology report, written by a radiologist, please rewrite it to be more explicit about the diagnostic decision reflected in the sentence, however, maintain the main meaning of the original sentence. A sentence is defined to be ambiguous because of (1) medical jargon with meanings different from everyday general usage, such as unremarkable; (2) contradictory findings in the same sentence; (3) misleading grammatical errors such as no period between full sentences. Now given a new sentence, answer me with its rewrite, without saying anything else. Sentence: aorta appears unchanged as compared to the study. Diagnostic: Abnormal. Rewrite:`
Returned Text: `\\... aorta appears unchanged as compared to the study.`

——————————————————————

In this unqualified example, the BioMed LM does not follow the instruction and just output an irrelevant sentence.

Prompt: `Here is a task to rewrite ambiguous sentences to be less ambiguous. Given a sentence in a radiology report, written by a radiologist, please rewrite it to be more explicit about the diagnostic decision reflected in the sentence, however, maintain the main meaning of the original sentence. A sentence is defined to be ambiguous because of (1) medical jargon with meanings different from everyday general usage, such as unremarkable; (2) contradictory findings in the same sentence; (3) misleading grammatical errors such as no period between full sentences. Now given a new sentence, answer me with its rewrite, without saying anything else. Sentence: apparent scarring within the lingula otherwise unremarkable. Diagnostic: Abnormal. Rewrite:`

Returned Text: `\\[Lingula\\] scarring.`

——————————————————————

**Document Level, NLU-Zero-shot**   In this task, we expect the output to have a prediction for every disease. However, BioMed LM just repeats the disease code as the output.

Prompt: `Here is a task to classify the observations of the chest reports. Given a radiology report written by a radiologist, please predict if every one of the following 14 observations is positive, negative, or unknown. The 14 observations are No Finding, Enlarged Cardiomediastinum, Cardiomegaly, Lung Lesion, Lung Opacity, Edema, Consolidation, Pneumonia, Atelectasis, Pneumothorax, Pleural Effusion, Pleural Other, Fracture, Support Devices. Answer me only with the tuples of each observation and its preidction, without saying anything else. There should be 14 tuples in your answer and tuples are separated by comma. Report: bibasilar changes in atelectatics. the lungs are otherwise clear. the cardiomedical silhouette and the hila are normal. right port-a-cath ends at the cavoatrial junction. there's no pneumothorax in this case. No pleural effusion. Mentioned observations:`
Returned Text: `No finding, Enlarged Cardiomediastinum, Cardiomegaly, Lung Lesion, Lung Opacity, Edema, Consolidation, Pneumonia, Atelectasis, Pneumothorax, Pleural Effusion, Pleural Other.`

**Document-Level, NLG-Two-shot**   We show an example of "freestyle" output from BioMed LM

Prompt: `Here is a summarization task. Given a radiology report written by a radiologist, please write a summary of the report. Here are 2 examples.`
`Report: clinical history 62-year-old male with bilateral bunions . please perform weighted views. a lateral weight bearing view of bilateral feet as well as a lateral non-weight bearing view of bilateral feet are studied. three views of the right foot show no fracture dislocation foreign body`

```
pathologic calcification or soft tissue
swelling.  the joint spaces are not
noticeable. there is a deformity of the
hallux valgus.  the angle of pitch is
within normal limits. three views of the
left foot show no fracture dislocation
foreign body pathologic calcification or
soft tissue swelling.  the joint spaces
are not noticeable. hallux valgus minor.
the angle of pitch is within normal
limits.  Summary:  1 bilateral hallux
valgus deformities.  2 no acute osseous
abnormalities. ssn7312ptc1job no. 1157.
Report:  there has been no significant
change in the patient's condition since
the patient's exam which was earlier in
the day.  the heart size is normal and
the lungs are free of disease.  on the
left base is again noted a small granulom.
Summary: for a active disease in the chest
there is no evidence.
Report: right and left feet have severe
hammertoes. calcaneus on the left shows
an achilles spur.  the severe hallux
valgus configuration of the feet is
observed.  a dominant finding was the
degree of luxation and deviation of first
metatarsal proximal phalangeal joint..
Summary:
Returned Text: 1 bilateral hammertoes.
2 no acute osseous abnormalities.
ssn7312ptc1job no. 1157.
Report: the patient has been seen in the
clinic for a routine checkup. The patient
has
```

in our Github repository):

## E   More Results

We considered popular LLMs such as GPT3, Chat-GPT, and Vicuna-7B (the instruction finetuned LLaMa) in the main context. But we also keep in mind that LLMs are rapidly developing and we are following up by adding the newest models into our evaluation. In Table 10, Table 11, Table 12, and Table 13, we provide more results with GPT4[10], LLaMa2 (Touvron et al., 2023b), LLaMa2-chat (Touvron et al., 2023b), GPT-NeoX citegpt-neox-20b, PMC-LLaMa (Wu et al., 2023), BioGPT (Luo et al., 2022) etc, which align with the findings already in the paper. We will keep working on follow-ups and add more evaluation results here (and also

---

[10]https://openai.com/gpt-4

| | Models | Chest | | Miscellaneous Domains | |
|---|---|---|---|---|---|
| | | Abnormality↑ | Ambiguity↑ | Abnormality↑ | Ambiguity↑ |
| Adapted PLMs with Fine-Tuning | zero-shot BioGPT | 0.7521 | 0.3012 | 0.6994 | 0.0543 |
| | few-shot BioGPT | 0.5966 | 0.2864 | 0.5990 | 0.0672 |
| | zero-shot PMC-LLaMa | 0.6663 | 0.2986 | 0.6189 | 0.0465 |
| | few-shot PMC-LLaMa | 0.7660 | 0.2606 | 0.7162 | 0.0574 |
| LLMs Prompted by Zero/Few Shot | zero-shot GPT4 | 0.9299 | 0.6596 | 0.9547 | 0.8462 |
| | few-shot GPT4 | 0.9145 | 0.7925 | 0.9417 | 0.8906 |
| | zero-shot GPT-NeoX | 0.6371 | 0.2747 | 0.5460 | 0.0651 |
| | few-shot GPT-NeoX | 0.5986 | 0.2578 | 0.5000 | 0.0648 |
| | zero-shot LLaMa2-chat | 0.4626 | 0.2864 | 0.4075 | 0.0679 |
| | few-shot LLaMa2-chat | 0.6551 | 0.2859 | 0.5685 | 0.0679 |
| | zero-shot LLaMa2 | 0.5568 | 0.3150 | 0.6474 | 0.0679 |
| | few-shot LLaMa2 | 0.4728 | 0.2864 | 0.4240 | 0.0679 |

Table 10: More results on sentence-level NLU tasks.

| | Models | Chest | | | Miscellaneous Domains | | |
|---|---|---|---|---|---|---|---|
| | | Disambiguation $\Delta Acc_{am} \uparrow$ | Content Distortion $\Delta Acc_{ab} \downarrow$ | BLEU4↑ | Disambiguation $\Delta Acc_{am} \uparrow$ | Content Distortion $\Delta Acc_{ab} \downarrow$ | BLEU4↑ |
| Adapted PLMs with Fine-Tuning | zero-shot BioGPT | 0.8184 | 0.2971 | 19.54 | 0.8623 | 0.4179 | 19.03 |
| | few-shot BioGPT | 0.8019 | 0.3036 | 22.43 | 0.8651 | 0.4382 | 22.32 |
| | zero-shot PMC-LLaMa | 0.8119 | 0.4389 | 19.03 | 0.9235 | 0.5393 | 18.77 |
| | few-shot PMC-LLaMa | 0.5577 | 0.2244 | 22.61 | 0.6719 | 0.3213 | 21.07 |
| LLMs Prompted by Zero/Few Shot | zero-shot GPT4 | 0.7921 | -0.0297 | 60.61 | 0.7562 | 0.1652 | 60.36 |
| | few-shot GPT4 | 0.7591 | 0.0000 | 63.66 | 0.7753 | 0.0697 | 67.67 |
| | zero-shot GPT-NeoX | 0.4125 | 0.0462 | 38.23 | 0.4786 | 0.2225 | 35.16 |
| | few-shot GPT-NeoX | -0.056 | 0.0396 | 24.65 | 0.0876 | 0.2517 | 25.91 |
| | zero-shot LLaMa2-chat | 0.4059 | 0.0033 | 53.23 | 0.3379 | 0.1314 | 53.20 |
| | few-shot LLaMa2-chat | 0.4521 | 0.0198 | 57.26 | 0.4480 | 0.1066 | 52.59 |
| | zero-shot LLaMa2 | 0.1320 | 0.0165 | 29.95 | 0.0921 | 0.0157 | 39.60 |
| | few-shot LLaMa2 | 0.1386 | 0.257 | 10.77 | 0.4044 | 0.1101 | 10.18 |

Table 11: More results on sentence-level NLG tasks.

| | Models | Chest | | Foot | | Ankle | |
|---|---|---|---|---|---|---|---|
| | | avg Accuracy↑ | avg EMR↑ | avg Accuracy↑ | avg EMR↑ | avg Accuracy↑ | avg EMR↑ |
| Adapted PLMs with Fine-Tuning | zero-shot BioGPT | 0.8276 | 0.0654 | 0.9222 | 0.0010 | 0.9299 | 0.0020 |
| | few-shot BioGPT | 0.7561 | 0.0030 | 0.9618 | 0.4196 | 0.9688 | 0.4860 |
| | zero-shot PMC-LLaMa | 0.8290 | 0.0663 | 0.9228 | 0.0010 | 0.9206 | 0.0010 |
| | few-shot PMC-LLaMa | 0.7549 | 0.0039 | 0.9604 | 0.4044 | 0.9228 | 0.0020 |
| LLMs Prompted by Zero/Few Shots | zero-shot GPT-NeoX | 0.8291 | 0.0663 | 0.9238 | 0.0033 | 0.9318 | 0.0194 |
| | few-shot GPT-NeoX | 0.7876 | 0.0537 | 0.9617 | 0.4186 | 0.9691 | 0.4908 |
| | zero-shot LLaMa2-chat | 0.3192 | 0.0294 | 0.9617 | 0.4186 | 0.9691 | 0.4908 |
| | few-shot LLaMa2-chat | 0.8192 | 0.0505 | 0.9617 | 0.4186 | 0.9691 | 0.4908 |
| | zero-shot LLaMa2 | 0.7995 | 0.0591 | 0.9518 | 0.3171 | 0.9669 | 0.4597 |
| | few-shot LLaMa2 | 0.8331 | 0.0722 | 0.9594 | 0.3898 | 0.9478 | 0.1135 |

Table 12: More results on document-level NLU tasks.

| Models | Miscellaneous Domains | | | | |
|---|---|---|---|---|---|
| | ROUGE-1↑ | ROUGE-2↑ | ROUGE-L↑ | Sum ↑ | BLEU4 ↑ |
| **Adapted PLMs with Fine-Tuning** | | | | | |
| zero-shot BioGPT | 18.1577 | 6.5828 | 16.9262 | 41.6667 | 8.10 |
| few-shot BioGPT | 12.3569 | 3.8041 | 11.6335 | 27.7945 | 5.26 |
| zero-shot PMC-LLaMa | 11.6813 | 3.8402 | 10.8875 | 26.409 | 7.52 |
| few-shot PMC-LLaMa | 14.5339 | 3.9377 | 13.0060 | 31.4776 | 8.46 |
| **LLMs Prompted by Zero/Few Shots** | | | | | |
| zero-shot GPT4 | 59.8081 | 34.6459 | 55.1671 | 149.6211 | 53.82 |
| few-shot GPT4 | 56.1813 | 31.6360 | 51.5054 | 139.3228 | 44.16 |
| zero-shot GPT-NeoX | 18.3383 | 6.6764 | 17.0308 | 42.0455 | 11.01 |
| few-shot GPT-NeoX | 12.3496 | 3.8196 | 11.6543 | 27.8235 | 8.11 |
| zero-shot LLaMa2-chat | 21.0446 | 6.9982 | 18.8494 | 46.8923 | 21.97 |
| few-shot LLaMa2-chat | 17.8764 | 5.6633 | 15.5410 | 39.0806 | 21.54 |
| zero-shot LLaMa2 | 18.0240 | 5.8142 | 16.5904 | 40.3801 | 15.37 |
| few-shot LLaMa2 | 21.4236 | 7.8157 | 20.0528 | 49.2921 | 18.92 |

Table 13: More results on document-level NLG tasks.