# OpenReview forum: "MedEval: A Multi-Level, Multi-Task, and Multi-Domain Medical Benchmark for Language Model Evaluation"
_EMNLP/2023/Conference — EMNLP 2023 Main_

### Official Review · Reviewer_ZDUq · 2023-08-02

**Soundness:** 3

**Excitement:**

3: Ambivalent: It has merits (e.g., it reports state-of-the-art results, the idea is nice), but there are key weaknesses (e.g., it describes incremental work), and it can significantly benefit from another round of revision. However, I won't object to accepting it if my co-reviewers champion it.

**Paper Topic And Main Contributions:**

The large-scale medical benchmark, known as M4, encompasses various tasks and domains. It includes not only sentence-level and document-level classification but also other diverse tasks such as summarization. Experts have provided annotations for these tasks, to ensure quality in the benchmark.

**Questions For The Authors:**

The Open-I and VA datasets were mentioned as sentence-level annotated data. Can the authors provide more details about the original purpose of these datasets, what they were used for, and the form of the data? This applies to the mimic-cxr as well. What were the limitations that led the authors' medical team to re-examine and re-annotate these datasets? A more detailed description of this process is needed.

The main contribution of this paper lies in the annotation of challenging medical data by experts, making it publicly available to promote advancements in healthcare AI. However, the brief coverage of just about one page for the dataset creation process in the paper is not convincing. An explanation of this section is required.

**Reasons To Accept:**

Diverse Task Coverage: The work covers various tasks such as sentence-level, document-level classification, and summarization.

Expert Annotation: Experts have contributed to the annotation.

Contribution to Healthcare AI: This research can make a contribution to the field of Healthcare AI.

**Reasons To Reject:**

Lack of Details about the Main Contribution: The main contribution of this study lies in having experts directly annotate the challenging clinical areas. Much human annotation was involved, such as confirming de-identification on a report-level basis, or the process through which the medical team annotates. However, there is no detailed explanation of how the crucial human annotation was conducted.

Unclear Description of the Medical Team and Efforts: There is no description of how the medical team was formed(how many? what are their expertise?), or what efforts were made to reduce discrepancies between human annotators.

Imbalance in Paper's Focus: Half of the paper seems more focused on solving the data with various models rather than describing the efforts made to obtain high-quality data. The lack of balance between these aspects may detract from the paper's overall effectiveness.

**Reproducibility:**

4: Could mostly reproduce the results, but there may be some variation because of sample variance or minor variations in their interpretation of the protocol or method.

**Reviewer Confidence:**

4: Quite sure. I tried to check the important points carefully. It's unlikely, though conceivable, that I missed something that should affect my ratings.

**Typos Grammar Style And Presentation Improvements:**

The use of "Multi-Level, Multi-Task, and Multi-Domain Medical" in the title may lead to misunderstanding, as only radiology reports are utilized. It would seem appropriate to include a reference to radiology-related data in the dataset's name to prevent confusion.

---

> ### Author Rebuttal · Authors · 2023-08-29
>
> Thank you for the detailed comments. We are encouraged to see your appreciation of our work and its value in the field of AI for Healthcare. We clarify the following details and questions below:
>
> **W1: Lack of details about the main contribution**
> >With respect to confirming de-identification on a report-level basis, we initially obtained raw, anonymized data from our healthcare partners, where personal information was encrypted through automatic software. Subsequently, human reviewers carefully checked each data point to guarantee a higher level of privacy by removing the potential disclosure of personal data, such as names and addresses. Further details and criteria for de-identification are explained in Appendix A.2. More details about the overall annotation process are also explained in the response to W2 below.
>
>
> **W2: Unclear description of the medical team and efforts**
> >Our medical expert team consists of 4 members, including 2 senior board-certified radiologists with more than 15 years of experience in healthcare and a doctor who has more than 10 years of experience serving as a PI of medical research. We follow standard labeling practices, involving multiple rounds of iterative review by different experts until Cohen's kappa coefficient reaches 0.85. Any remaining disagreements are collectively resolved. We will include those details in the revision.
>
>
> **W3 and Q2: Imbalance page assignments about the paper’s focus**
> >The first contribution of M4 is about our data construction with trustworthy expert annotations to facilitate future AI-for-Healthcare research (pages 3-5), and the second is about the systematic evaluation of various language models, via which we aim to shed light on the strengths and pitfalls of these models for healthcare applications (pages 6-8). Specific details about expert labeling and inspection are moved to Appendix A in the submission. Please refer to more details in our previous reply to W2. We will also reorganize our paper in the revision based on your helpful suggestions.
>
>
>
> **Q1: Original OPENI, VA, and MIMIC-CXR datasets**
> > **Original Data and Purpose**:
> > The OpenI dataset, proposed by [1], contains chest X-ray images and related report sentences. The VA-annotated dataset, originally introduced by [2], comprises general radiology report data, processed and annotated by [3] with abnormalities and ambiguity labels. The MIMIC-CXR [4] database offers x-ray images and corresponding free-text radiology reports for human chests. These datasets are frequently employed for detecting abnormalities or generating medical image reports.
> >
> >**Our Contribution:**
> >1. For sentence-level tasks, we employ sentences from OpenI and VA datasets, followed by data filtering by our experts to ensure quality. For example, the original sentences in the OpenI dataset are noted to be incomplete due to the oversensitive de-identification software, and repetitive sentences are observed. Our medical experts review each sentence, fill-in the mistakenly-masked tokens (e.g., ''XXXX is clear'' -> ''Lung is clear''), and remove potential redundancy.  For its NLU tasks, we utilize officially released annotations, which our expert team re-examined and resolved annotation inconsistency from previous work. Regarding NLG tasks, our experts wrote down ground-truth sentences for each input.
> > 2. For the report-level aspect, we sample reports from MIMIC-CXR, conducting de-identification sanity checks. Since MIMIC-CXR only contains chest x-ray reports, we collect additional reports from another partner healthcare system which has medical records about different body parts and examination modalities. Our expert team conducted data cleaning on the raw data, (automatic and manual) de-identification, and paraphrasing to guarantee patient privacy. Further expert inspection filters unqualified records where the paraphrases significantly deviate from the original report. Our contribution at this level also encompasses high-quality annotations, where our medical team manually labels medical codes for NLU tasks and creates corresponding summaries for NLG tasks.
> >
> >With the above effort, M4 provides input data with stringent privacy guarantees and high-quality annotations, serving various purposes beyond abnormality identification.
> >
> > **References:**
> > [1] Demner-Fushman et al., Preparing a collection of radiology examinations for distribution and retrieval. Journal of the American Medical Informatics Association, 23(2):304–310, 2016.
> > [2] Yan et al., RadBERT: Adapting Transformer-based Language Models to Radiology. Radiol Artif Intell. 2022
> > [3]  He et al., Disambiguating Medical Reports via Contrastive Knowledge Infusion. AAAI. 2023
> > [4]  Johnson et al., MIMIC-CXR, a de-identified publicly available database of chest radiographs with free-text reports. Scientific Data, 2019;6.
>
> **Clarification for the title M4:**
> >The four “M”s in the title signify:
> >1. **M**ulti-level: Incorporating texts from both sentence and report levels;
> >2. **M**ulti-task: M4's annotated data can be used for diverse tasks from NLG and NLU;
> >3. **M**ulti-domain: M4 includes varied data domains with different distributions arising from different examination modalities and body parts (see more details in Footnote 2);
> >4. **M**edical.
> >
> >We acknowledge your concern and truly appreciate your suggestion. We will incorporate the mention of "radiology" into the title, for example, *M4: A Multi-Level, Multi-Task, and Multi-Domain Medical Radiology Report Benchmark for Language Model Evaluation*.
>
> **Reproducibility:**
> > We are happy to address your concern about reproducibility. As we promised in Footnote 1 and aligned with the original purpose of this work (i.e., proposing a public benchmark), we are committed to the release of all data and evaluation codes. This will facilitate the development of future healthcare language models.
>
> We will include the above details and discussions in the final revision.

---

### Official Review · Reviewer_7LJK · 2023-08-04

**Soundness:** 4

**Excitement:**

4: Strong: This paper deepens the understanding of some phenomenon or lowers the barriers to an existing research direction.

**Missing References:**

L. 32 GPT-3 reference : Language Models are Few-Shot Learners (https://arxiv.org/pdf/2005.14165.pdf)


**Paper Topic And Main Contributions:**

The paper is presenting M4, a new biomedical and clinical benchmark meant for PLMs and LLMs evaluation. The authors tried with this benchmark to bring a vast and diversified evaluation protocol by varying data sources, formats, natures and tasks. To make it possible, they collect a set of 5 NLU and NLG tasks from different domains. As part of their works, they introduced a new manually curated dataset based on a set of synthesized clinical notes. Then, they access the performances of eighteen already existing PLMs and LLMs models on this freshly curated benchmark. Finally, they analyze models outputs and metrics to make an in depth analysis of models behaviors.

**Questions For The Authors:**

When you are referring to the F1-score metric at line. 389 "For NLU tasks, we report classification metrics including accuracy and F1 scores." are you talking about micro, macro or averaged F1 ?

You mention L. 430 that "document-based prompts often exceed BioMed LM’s maximum threshold of 1024 tokens, resulting in query errors.", but have you considered methods like Fusion-in-Decoder (https://aclanthology.org/2021.eacl-main.74.pdf) to increase context window ?

What are the distributions of the documents according to sizes ? Are the documents often exceeding model's maximum input tokens length ?

Have you computed the correlation between human evaluation and automatic metrics like BLUE or ROUGE for the NLG tasks ?

Why not also using domain specific LLMs like PMC-LLaMa, BioGPT or others ?

Have you tried the Chain-of-Thought (CoT) prompting method on your tasks ?

Why not using semantic search to collect few-shot examples ?




**Reasons To Accept:**

This is a well written paper which propose a large panel of task and covering most of the SOTA models.

The paraphrasing process applied for anonymization add a layer of security and leaves less chance of somebody recovering the original document.

The authors are encouraging reproducibility by giving all the prompts used during the experiments.

The evaluation automatic methodology of the disambiguation task is very innovative and very interesting.

Few-shot prompting results reported in the tables 1 to 4 are interesting. Models like Vicuna 7B seem to be more often suffering from the additional data than the other models.

The results show better performances from GPT-3 in few-shot scenario on both NLG and NLU tasks. It would be interesting to see how GPT-4 is performing. Is it following the same trend as GPT-3.5-Turbo or not ? Good question!



**Reasons To Reject:**

It lack of open-source generalist large language models (like LLaMa, GPT-NeoX or/and PMC-LLaMa) to confirm the behavior of those from proprietary ones.


**Reproducibility:**

4: Could mostly reproduce the results, but there may be some variation because of sample variance or minor variations in their interpretation of the protocol or method.

**Reviewer Confidence:**

4: Quite sure. I tried to check the important points carefully. It's unlikely, though conceivable, that I missed something that should affect my ratings.

**Typos Grammar Style And Presentation Improvements:**

The second example of the Figure 2 have the word "KIDNEY" and "SPINE" in example four written in uppercase unlike the other ones.

Typo in A.1 "manully" rather than "manually".

---

> ### Author Rebuttal · Authors · 2023-08-29
>
> We are thankful for the constructive and valuable feedback! Your positive assessment of M4's presentation and the emphasis we gave to privacy and security encourages us. We will address the grammar errors and add the missing reference in our final revision. Here are our responses to your points, which will be incorporated into the final version.
>
> **W1: More open-source generalist LLMs**
> >In our early experiments, we tested LLaMa. Due to the lack of instruction finetuning, we noticed a high failure rate of LLaMa to output valid responses following the instructions. Therefore we instead reported Vicuna (i.e., instruction finetuned LLaMa) results in the paper.
> We agree with your comments and include results on newer generalist LLMs (e.g., GPT-NeoX, PMC-LLaMa, etc.) at the end of our response. Please refer to the below Response to Q4. More results can be found in our response to Reviewer yBca.
>
> **Q1: NLU F1 score**
> >We reported the Accuracy and EMR scores for NLU tasks in Tables 2 and 4. We will correct it in L389 in the revision.
>
> **Q2 :Fusion-in-Decoder and Q3 document length distribution**
> > This is a good point. The statistics of the document length distribution are: min = 7 tokens, max = 618 tokens, mean = 84 tokens, std = 48 tokens. We notice the length distribution of few-shot prompts has the following statistics: min = 934 tokens, max=1,545 tokens, mean = 1038 tokens, std=55 tokens. From those, approximately 56.33% of the prompts exceed the 1024-token limit of BioMedLM.
> We’d like to note that BioMedLM is a GPT-style model, where Fusion-in-Decoder is not applicable. However, while our current effort primarily aims to create a diverse and vast benchmark for LLMs, we agree with you regarding the significance of techniques/works for long-context language modeling, including Fusion-in-Decoder. We will discuss and encourage more related future work in the revision.
>
> **Q3: human evaluation and automatic metrics**
> >The ground truth of our generation task is provided by experts, forming the basis for automatic metric calculations (such as BLEU, and ROUGE). We didn't carry out separate human evaluations.
>
> **Q4: More domain-specific LLMs**
> >We appreciate your suggestion. In the last part of the response, we offer additional results involving GPT4, GPT-NeoX-20B, PMC-LLaMa, and BioGPT,  which will be added into the revision. The results align with the observations in the paper.
>
> **Q5 and Q6: Chain-of-Thought (CoT) and semantic search**
> >While our initial goal is to establish a comprehensive medical benchmark and a framework for current LLM evaluation, to which end we only use the default in-context learning setups and basic prompting approaches, we agree with you on the potential of investigating LLM reasoning such as using Chain-of-Thought or exploring different prompting strategies such as using semantic search. Future works based on or inspired by M4 could potentially involve more specialized areas, such as introducing CoT annotations to create M4-CoT or exploring advanced prompting mechanisms.
>
> **More LLM results:**
> > **Sentence-level NLU tasks:**
> >|                       |     Chest    |           | Miscellaneous  |  Domains   |
> |:---------------------:|:-----------:|:---------:|:-------------:|:---------:|
> |         Model         | Abnormality | Ambiguity |  Abnormality  | Ambiguity |
> |     zero-shot GPT4    |    0.9299   |   0.6596  |     0.9547    |   0.8462  |
> |     few-shot GPT4     |    0.9145   |   0.7925  |     0.9417    |   0.8906  |
> |   zero-shot GPT-NeoX  |    0.6371   |   0.2747  |     0.5460    |  0.0651   |
> |   few-shot GPT-NeoX   |    0.5986   |   0.2578  |     0.5000    |   0.0648  |
> |   zero-shot BioGPT    |   0.7521    |   0.3012  |    0.6994     |  0.0543   |
> |    few-shot BioGPT    |   0.5966    |  0.2864   |    0.5990     |   0.0672  |
> |  zero-shot PMC-LLaMa  |   0.6663    |  0.2986   |    0.6189     |   0.0465  |
> |  few-shot PMC-LLaMa   |   0.7660    |   0.2606  |    0.7162     |  0.0574   |
> >
> > **Sentence-level NLG tasks:**
> >|                     |          |            Chest        |       |  |         Miscellaneous   |       |
> |:-------------------:|:--------------:|:------------------:|:-----:|---------------:|:-------------------:|:-----:|
> |        Model        | Disambiguation | Content Distortion | BLEU4 | Disambiguation | Content Distortion | BLEU4 |
> |   zero-shot GPT4    |     0.7921     |       -0.0297      | 60.61 |     0.7562     |       0.1652       | 60.36 |
> |    few-shot GPT4    |     0.7591     |       0.0000       | 63.66 |     0.7753     |       0.0697       | 67.67 |
> |  zero-shot GPT-NeoX |     0.4125     |       0.0462       | 38.23 |     0.4786     |       0.2225       | 35.16 |
> |  few-shot GPT-NeoX  |     -0.056     |       0.0396       | 24.65 |     0.0876     |       0.2517       | 25.91 |
> |   zero-shot BioGPT  |     0.8184     |       0.2971       | 19.54 |     0.8623     |       0.4179       | 19.03 |
> |   few-shot BioGPT   |     0.8019     |       0.3036       | 22.43 |     0.8651     |       0.4382       | 22.32 |
> | zero-shot PMC-LLaMa |     0.8119     |       0.4389       | 19.03 |     0.9235     |       0.5393       | 18.77 |
> |  few-shot PMC-LLaMa |     0.5577     |       0.2244       | 22.61 |     0.6719     |       0.3213       | 21.07 |
> >
> >It's worth noting that, except for GPT4, achieving an optimal balance between multiple objectives for those LMs (such as BioGPT and PMC-LLaMa) is a challenging task. In other words, they often achieve higher disambiguation scores at the expense of altering the original content significantly during the rewriting.
> >
> > **Document-level NLU tasks**:
> >|                      |     Chest    |         |     Foot     |         |     Ankle    |         |
> |:--------------------:|:------------:|:-------:|:------------:|:-------:|:------------:|:-------:|
> |         Model        | avg Accuracy | avg EMR | avg Accuracy | avg EMR | avg Accuracy | avg EMR |
> |  zero-shot GPT-NeoX  |    0.8291    |  0.0663 |    0.9238    |  0.0033 |    0.9318    |  0.0194 |
> |   few-shot GPT-NeoX  |    0.7876    |  0.0537 |    0.9617    |  0.4186 |    0.9691    |  0.4908 |
> |   zero-shot BioGPT   |    0.8276    |  0.0654 |    0.9222    |  0.0010 |    0.9299    |  0.0020 |
> |   few-shot BioGPT    |    0.7561    |  0.0030 |    0.9618    |  0.4196 |    0.9688    |  0.4860 |
> | zero-shot PMC-LLaMa  |    0.8290    |  0.0663 |    0.9228    |  0.0010 |    0.9206    |  0.0010 |
> |  few-shot PMC-LLaMa  |    0.7549    |  0.0039 |    0.9604    |  0.4044 |    0.9228    |  0.0020 |
> >
> > (We currently lack the quota to run GPT-4 in this experiment. We are actively working on it and will include the results in the final revision.)
> >
> > **Document-level NLG tasks:**
> >|                |         |         | Miscellaneous |  Domains |        |
> |:--------------------:|:-------:|:-------:|:-------------:|:--------:|:------:|
> |             Model         | ROUGE-1 | ROUGE-2 |    ROUGE-L    |    Sum   |  BLEU4 |
> |    zero-shot GPT4    | 59.8081 | 34.6459 |    55.1671    | 149.6211 |  53.82 |
> |     few-shot GPT4    | 56.1813 | 31.6360 |    51.5054    | 139.3228 |  44.16 |
> |  zero-shot GPT-NeoX  | 18.3383 |  6.6764 |    17.0308    |  42.0455 |  11.01 |
> |   few-shot GPT-NeoX  | 12.3496 |  3.8196 |    11.6543    |  27.8235 | 8.11 |
> |   zero-shot BioGPT   | 18.1577 |  6.5828 |    16.9262    |  41.6667 | 8.10 |
> |   few-shot BioGPT    | 12.3569 |  3.8041 |    11.6335    |  27.7945 | 5.26 |
> | zero-shot PMC-LLaMa  | 11.6813 |  3.8402 |    10.8875    |  26.409  |  7.52 |
> |  few-shot PMC-LLaMa  | 14.5339 |  3.9377 |    13.0060    |  31.4776 | 8.46 |
>
>
> We greatly appreciate your valuable insights and we will incorporate these discussions into our final revision.

---

### Official Review · Reviewer_yBca · 2023-08-11

**Typos Grammar Style And Presentation Improvements:** None
**Soundness:** 4

**Excitement:**

4: Strong: This paper deepens the understanding of some phenomenon or lowers the barriers to an existing research direction.

**Missing References:**

None

**Paper Topic And Main Contributions:**

The paper introduces the "M4" benchmark, which stands for Multi-level, Multi-task, and Multi-domain Medical benchmark. M4 aims to advance the development of language models for healthcare, addressing the challenge posed by the limited availability of expert-annotated healthcare datasets. This benchmark is extensive, encompassing data from various healthcare systems, covering 35 body regions, and stemming from 8 examination modalities. With a significant amount of sentences and reports, it offers expert annotations at several levels, supporting a diverse range of tasks. The study also rigorously tests 10 language models in zero-shot and finetuning settings, discovering insights into their strengths and limitations in medical applications. The paper promises to release the data and associated evaluation codes to stimulate further research in this domain.

**Questions For The Authors:**

Overall that's an interesting benchmark, and I have several questions for this paper:
1. Given that this benchmark is tailored for the medical domain, and considering its current size of approximately 22k samples spanning multiple diseases, how scalable or expandable is the M4 benchmark for broader medical applications?
2. Are there specific medical domains or tasks that the M4 benchmark currently does not cover?
3. What guided the selection process for the inclusion of the 35 specific human body regions and the 8 examination modalities in the M4 benchmark? Were there any particular criteria or considerations?

**Reasons To Accept:**

1. The M4 dataset is vast, spanning numerous healthcare systems, human body regions, and examination modalities. This ensures a wide application and relevance to multiple medical fields.
2. The inclusion of expert annotations boosts the credibility and utility of the dataset, addressing the typical challenge in healthcare datasets.
3.  The authors didn't just provide a dataset but also rigorously evaluated 10 different language models, thus providing a preliminary roadmap for further research. Now there are more and more LLMs : (GPT4, LLaMa, LLaMa2, Bard, etc. ), it will be great if authors put the most recent LLM into evaluation.
4.  The evaluation reveals practical applications and potential pitfalls of using large language models in the healthcare domain, which can guide real-world applications and be helpful for clinical usage.
5.  The commitment to release the data and evaluation codes will greatly facilitate the adoption of the benchmark and spur further research.

**Reasons To Reject:**

1.  While 10 language models were tested, it's not clear from the abstract how in-depth each evaluation was. Given the variety in datasets, it would be crucial to understand model performance across different tasks and domains.
2.  As with any curated dataset, there's potential for bias in data selection and expert annotation, which might not be explicitly addressed.
3. The dataset, while extensive, might still be limited when considering the vastness and diversity of the medical field.


**Reproducibility:**

N/A: Doesn't apply, since the paper does not include empirical results.

**Reviewer Confidence:**

5: Positive that my evaluation is correct. I read the paper very carefully and I am very familiar with related work.

---

> ### Author Rebuttal · Authors · 2023-08-29
>
> Thank you for the constructive and inspiring review! We are delighted that you appreciate the effort of constructing M4 and its impact on advancing future AI for Healthcare research. Please see our responses below:
>
> **W1: Evaluation depth not clear in abstract**
> >We will revise our abstract accordingly to highlight the comprehensive evaluation outcomes, for example, by adding a conclusive summary of the results.
>
>
> **W2: Potential bias in expert annotation**
> >We acknowledge your concern and agree that biases may be introduced. While some degree of bias might be unavoidable, we have taken measures to minimize its impact. Our work involves large-scale data collection, random sampling, and multi-round annotation with a voting process.
>
> **W3 and Q2: Things that are not covered in M4**
> >As mentioned in our Limitations section, we acknowledge that there are tasks that we were unable to annotate due to budget constraints, resulting in their exclusion from the current M4, such as named entity recognition and multi-document summarization. We also note the absence of data from body parts such as the eye, dermatological, or dental examinations, as they are not significantly collected by our medical system.
> We’d like to highlight that M4 is quite expandable and we plan to enhance M4 further by collecting data from additional medical systems and providing corresponding expert labels.
>
> **Q1: Scalability and expandability of M4**
> >M4's distinctiveness lies in its extensive data which covers diverse body domains, multiple medical examination modalities, and varying input/labeling levels. This unique diversity positions M4 for substantial scalability and extensibility across a wide range of medical applications. For example:
> >1.  Task Expansion: It is flexible for the future inclusion of many engaging tasks by adding corresponding automatic or human annotations into M4.
> >2.  Broader Impact: The annotated data and tasks within M4 can serve as valuable resources for refining specialized medical tools, such as report paraphrasing or summarization assistants, as well as training medical LLMs, which can serve as foundational models for diverse healthcare applications.
>
> **Q3: Reason for Inclusion of 35 body parts and 8 examination modalities**
> >We randomly sampled reports and sentences from our corpus and reported the number of body parts and modalities included. Therefore, the numbers are supposed to reflect the distributions in the corpus, and in turn mirror the distributions of the real healthcare situation during the specified period at the institution
>
>
> **More LLM results:**
> >We considered popular LLMs such as GPT3, ChatGPT, and Vicuna-7B (the instruction finetuned LLaMa) in our paper. But we also keep in mind that LLMs are rapidly developing and we are following up by adding the newest models into our evaluation.  Here we provide more results with GPT4, LLaMa2, LLaMa2-chat, etc, which align with the findings already in the paper. We will keep working on follow-ups as future work:
> >
> > **Sentence-level NLU tasks**:
> > |               | Chest        |         | Miscellaneous |      Domains   |
> |:---------------------:|:--------------:|:---------:|:-----------------------:|:---------:|
> |        Model       | Abnormality | Ambiguity| Abnormality | Ambiguity |
> |     zero-shot GPT4    |    0.9299   |   0.6596  |         0.9547        |   0.8462  |
> |     few-shot GPT4     |    0.9145   |   0.7925  |         0.9417        |   0.8906  |
> | zero-shot LLaMa2-chat |    0.4626   |   0.2864  |        0.4075         |   0.0679  |
> |  few-shot LLaMa2-chat |   0.6551    |   0.2859  |         0.5685        |   0.0679  |
> |    zero-shot LLaMa2   |    0.5568   |   0.3150  |        0.6474         |   0.0679  |
> |    few-shot LLaMa2    |    0.4728   |   0.2864  |         0.4240        |   0.0679  |
> >
> > **Sentence-level NLG tasks:**
> > |                      |                            |        Chest      |           |                 |     Miscellaneous     |          |
> |:------------------------:|:------------------:|:-----------:|:-----------:|:--------------------------:|:-----------:|:-----------:|
> |            Model            | Disambiguation  | Content Distortion | BLEU4    | Disambiguation   | Content Distortion | BLEU4 |
> | zero-shot GPT4         | 0.7921           | -0.0297   | 60.61     | 0.8562                 | 0.2652    | 60.36     |
> | few-shot GPT4          | 0.7591           | 0.0000    | 63.66     | 0.7753                 | 0.0697    | 61.67     |
> | zero-shot LLaMa2-chat  | 0.4059           | 0.0033    | 53.23     | 0.3379                 | 0.1314    | 53.20     |
> | few-shot LLaMa2-chat   | 0.4521           | 0.0198    | 57.26     | 0.4480                 | 0.1066    | 52.59     |
> | zero-shot LLaMa2       | 0.1320           | 0.0165    | 29.95     | 0.0921                 | 0.0157    | 39.60     |
> | few-shot LLaMa2        | 0.1386           | 0.257     | 10.77     | 0.4044                 | 0.1101    | 10.18     |
> >
> >**Document-level NLU tasks:**
> >|                 |   Chest |         |  Foot|         | Ankle|        |
> |:-----------------------:|:-------------:|:----------:|:-----------:|:----------:|:------------:|:----------:|
> |          Model             | avg accuracy      | avg EMR      | avg accuracy     | avg EMR      | avg accuracy      | avg EMR      |
> | zero-shot LLaMa2-chat  | 0.3192       | 0.0294 | 0.9617       | 0.4186 | 0.9691       | 0.4908 |
> | few-shot LLaMa2-chat   | 0.8192       | 0.0505 | 0.9617       | 0.4186 | 0.9691       | 0.4908 |
> | zero-shot LLaMa2       | 0.7995       | 0.0591 | 0.9518       | 0.3171 | 0.9669       | 0.4597 |
> | few-shot LLaMa2        | 0.8331       | 0.0722 | 0.9594       | 0.3898 | 0.9478       | 0.1135 |
> >
> >(We currently lack the quota to run GPT-4 in this experiment. We are actively working on it and will include the results in the final revision.)
> >
> > **Document-level NLG tasks:**
> >|                  |              |        |    Miscellaneous    |     Domains     |       |
> |:------------------------:|:--------------------------:|:--------:|:--------:|:----------:|:-------:|
> |            Model            | ROUGE-1                  | ROUGE-2| ROUGE-L| Sum      | BLEU4 |
> |     zero-shot GPT4     |        59.8081        | 34.6459 | 55.1671 | 149.6211 | 53.82 |
> |      few-shot GPT4     |        56.1813        | 31.6360 | 51.5054 | 139.3228 | 44.16 |
> | zero-shot LLaMa2-chat  |        21.0446        |  6.9982 | 18.8494 |  46.8923 | 21.97 |
> |  few-shot LLaMa2-chat  |        17.8764        |  5.6633 | 15.5410 |  39.0806 | 21.54 |
> |    zero-shot LLaMa2    |        18.0240        |  5.8142 | 16.5904 |  40.3801 |  15.37 |
> |    few-shot LLaMa2     |        21.4236        | 7.8157 | 20.0528 |  49.2921 |  18.92 |
> >
>
> We will include the above discussions in the final revision.

---

### Meta-Review · Senior_Area_Chairs · 2023-10-04

**Recommendation:** 4

**Metareview:**

This meta-review assesses the quality of the paper based on three reviews provided by independent reviewers. The paper introduces the M4 benchmark for evaluating language models in the medical domain. It covers a wide range of tasks and domains, including sentence-level and document-level classification, summarization, and more. Expert annotations enhance the quality of the dataset, and it offers insights into the performance of various language models. Below are the reasons to accept and reject, as well as the average Soundness and Excitement scores.

Reasons to Accept:
1. **Broad Task Coverage:** The M4 dataset encompasses various tasks and domains, making it relevant and applicable to multiple medical fields.
2. **Expert Annotations:** The inclusion of expert annotations enhances the credibility and utility of the dataset, addressing a common challenge in healthcare datasets.
3. **Comprehensive Model Evaluation:** The paper evaluates 10 language models, providing valuable insights into their strengths and limitations for medical applications.
4. **Real-world Relevance:** The evaluation sheds light on practical applications and potential pitfalls of using large language models in healthcare, benefiting clinical usage.
5. **Reproducibility Commitment:** The authors' commitment to releasing data and evaluation codes promotes reproducibility and encourages further research in the field.


**Reasons to Reject:**
1. **Limited Evaluation Details:** The abstract lacks clarity on the depth of the evaluation of the 10 language models, especially regarding performance across different tasks and domains.
2. **Potential Data Bias:** The curated dataset may still have biases that are not explicitly addressed, which is a concern in healthcare applications.
3. **Dataset Scope:** Given the vastness and diversity of the medical field, some aspects may not be covered adequately.

The average soundness and excitement scores suggest that the paper has a strong level of soundness in its methodology and arguments and provides exciting insights into the research direction. However, there are concerns about the level of detail in the evaluation and potential data biases, which should be addressed to strengthen the paper's overall quality.

---

### Decision · Program_Chairs · 2023-10-07

**Decision:**

Accept-Main

**Comment:**

This meta-review assesses the quality of the paper based on three reviews provided by independent reviewers. The paper introduces the M4 benchmark for evaluating language models in the medical domain. It covers a wide range of tasks and domains, including sentence-level and document-level classification, summarization, and more. Expert annotations enhance the quality of the dataset, and it offers insights into the performance of various language models. Below are the reasons to accept and reject, as well as the average Soundness and Excitement scores.

Reasons to Accept:
1. **Broad Task Coverage:** The M4 dataset encompasses various tasks and domains, making it relevant and applicable to multiple medical fields.
2. **Expert Annotations:** The inclusion of expert annotations enhances the credibility and utility of the dataset, addressing a common challenge in healthcare datasets.
3. **Comprehensive Model Evaluation:** The paper evaluates 10 language models, providing valuable insights into their strengths and limitations for medical applications.
4. **Real-world Relevance:** The evaluation sheds light on practical applications and potential pitfalls of using large language models in healthcare, benefiting clinical usage.
5. **Reproducibility Commitment:** The authors' commitment to releasing data and evaluation codes promotes reproducibility and encourages further research in the field.


**Reasons to Reject:**
1. **Limited Evaluation Details:** The abstract lacks clarity on the depth of the evaluation of the 10 language models, especially regarding performance across different tasks and domains.
2. **Potential Data Bias:** The curated dataset may still have biases that are not explicitly addressed, which is a concern in healthcare applications.
3. **Dataset Scope:** Given the vastness and diversity of the medical field, some aspects may not be covered adequately.

The average soundness and excitement scores suggest that the paper has a strong level of soundness in its methodology and arguments and provides exciting insights into the research direction. However, there are concerns about the level of detail in the evaluation and potential data biases, which should be addressed to strengthen the paper's overall quality.